

# Flash visual evoked potentials in diurnal birds of prey

Maurizio Dondi, Fabio Biaggi, Francesco Di Ianni, Pier Luigi Dodi and Fausto Quintavalla

Department of Veterinary Science, University of Parma, Parma, Italy

## ABSTRACT

The objective of this pilot study was to evaluate the feasibility of Flash Visual Evoked Potentials (FVEPs) testing in birds of prey in a clinical setting and to describe the protocol and the baseline data for normal vision in this species. FVEP recordings were obtained from 6 normal adult birds of prey: n. 2 Harris's Hawks (*Parabuteo unicinctus*), n. 1 Lanner Falcon (*Falco biarmicus*), n. 2 Gyrfalcons (*Falco rusticolus*) and n. 1 Saker Falcon (*Falco cherrug*). Before carrying out VEP tests, all animals underwent neurologic and ophthalmic routine examination. Waveforms were analysed to identify reproducible peaks from random variation of baseline. At least three positive and negative peaks were highlighted in all tracks with elevated repeatability. Measurements consisted of the absolute and relative latencies of these peaks (P1, N1, P2, N2, P3, and N3) and their peak-to-peak amplitudes. Both the peak latency and wave morphology achieved from normal animals were similar to those obtained previously in other animal species. This test can be easily and safely performed in a clinical setting in birds of prey and could be useful for an objective assessment of visual function.

## INTRODUCTION

The relationship between man and birds of prey has been known since ancient times. In fact, it is believed that falconry originated from a hunting technique used on the Mongolian plateau around 6000 B.C. Over the millennia, the art of falconry spread all around the world, becoming a well-known form of huntsmanship practised by the noble classes (*Frederici II*, *1260*). Although certain pathologies affecting birds of prey have been known for centuries, anatomical and clinical data available in the literature remain scarce, even though over the past years veterinary interest towards birds of prey has significantly grown (*Redig*, *1993*; *Farrington*, *2004*; *Zucca*, *2004*; *Cooper*, *2004*).

In recent years, an increase in public awareness on environmental protection and integrated catchment management has led to a high demand in specialized diagnostic services with the creation of veterinary centres and rehabilitation facilities also dedicated to birds of prey (*Tristan*, *2010*). In these facilities, the most commonly encountered medical conditions are structural eye pathologies (ophthalmic trauma) with a prevalence between 28% (*Murphy*, *1987*) and 48% (*Labelle et al.*, *2012*). The

Corresponding author
Maurizio Dondi,
maurizio.dondi@unipr.it

consequence of irreversible damage to the sight of these animals is extremely relevant with regards to survival. In fact, in the above studies, only 12% of animals can be freed, whilst 43% must undergo euthanasia. The remaining 45% of animals is admitted to prolonged recovery centres (*Tristan*, *2010*). This is due to the fact that in birds of prey, sight is of fundamental importance in order to maintain predatory skills. A reduction in visual acuity or loss of stereoscopic sight following partial bilateral or complete unilateral visual lesions can determine reduced survival in nature or a reduction of their use in falconry (*Labelle et al.*, *2012*).

Optimal sight is determined by the correct functioning of all the anatomical structures that constitute the visual pathways, from the eye to the wustl and entopallium, and its assessment in birds of prey requires an articulate clinical and instrumental approach. The first approach, which consists in observing the animal in the aviary, allows to assess the bird's ability to avoid objects, as well as its predatory technique. This approach is limited as it does not allow to identify mild visual impairment. These visual defect are compensated in captivity, but not enable animals to survive in their natural habitat (*Pauli et al.*, *2007*).

The second approach is represented by ophthalmic examination. On the one hand, this examination allows us to accurately identify traumatic and inflammatory lesions of the eyes. However, on the other, it does not provide functional data especially on post-retinal visual pathways. In today's clinical practice, the functional assessment of the post-retinal visual pathways is based exclusively on cranial nerve examination. Whilst providing functional data, this examination is not particularly sensitive or objective. In contrast, instrumental tests, such as electroretinography (ERG) and Visual Evoked Potentials (VEP), provide objective and quantitative data on the functionality of the retina and of the post-retinal visual pathways (*Roze, Lucciani & Auphan*, *1990*; *Willis & Wilkie*, *1999*; *Clippinger, Bennett & Platt*, *2007*; *Labelle et al.*, *2012*).

ERG provides objective functional data on the retina and is widely used in most animal species of veterinary interest. Over the past years, this test has been used by veterinary as a routine test to be carried out on birds of prey before releasing them back into their natural environment (*Narfström et al.*, *2002*; *Labelle et al.*, *2012*).

The functional assessment of the post-retinal visual pathways has long been carried out in humans and in dogs using Visual Evoked Potential Testing. Unlike ERG, VEP testing provides functional information mainly with regards to lesions of the optic nerve and of the central visual pathways (*Bichsel et al.*, *1988*; *Sims et al.*, *1989*; *Strain, Jackson & Tedford*, *1990*; *Kimotsuki et al.*, *2005a*; *Kimotsuki et al.*, *2005b*; *Itoh et al.*, *2010*). In birds the use of visual evoked potentials was carried out with invasive techniques, with the aim of determining the origin of the potential of the individual structures of the visual pathways. As experimental animals were used pigeons and zebra finch (*Parker & Deltus*, *1972*; *Engelage & Bischof*, *1989*, *Engelage & Bischof*, *1988*; *Bredenkotter & Bischof*, *1990*; *Engelage & Bischof*, *1990*; *Wu, McGoogan & Cassone*, *2000*, *Bredenkotter & Bischof*, *2003*). Searching the veterinary literature revealed that VEP test is not yet used in raptors with clinical purposes, and that protocols and normal reference values are lacking for these species. Visual evoked potentials (VEPs) are electro-diagnostic tests, which allow us to study the activation of visual
pathways, from the retina to cortical areas, as a result of light stimulation. The activations of these neuro-anatomic structures are represented, on the recorded waveforms, as a series of waves characterized by positive and negative peaks representing the variation of the electric field over time (*Bichsel et al.*, *1988*; *Sims et al.*, *1989*; *Strain, Jackson & Tedford*, *1990*; *Kimotsuki et al.*, *2005a*; *Kimotsuki et al.*, *2005b*; *Itoh et al.*, *2010*).

The objective of this pilot study was to evaluate the feasibility of VEP testing in birds of prey in a clinical setting and to describe a routine method to define baseline data for normal vision in diurnal birds of prey (*Thabane et al.*, *2010*).

## METHODS

### Animals

FVEP recordings were obtained from the right ($N = 5$) and left ($N = 6$) eyes of 6 normal adult birds of prey: Harris's Hawks ($N = 2$) (*Parabuteo unicinctus*), Lanner Falcon ($N = 1$) (*Falco biarmicus*), Gyrfalcons ($N = 2$) (*Falco rusticolus*) and Saker Falcon ($N = 1$) (*Falco cherrug*). The data on VEP responses were collected during regular routine check-ups carried out to assess health and hunting predisposition in a population of client-owned birds of prey used in falconry at the Veterinary Hospital of the University of Parma (Italy) in the year 2013. Owner consent was obtained from all the participants of the study after having thoroughly informed the owner about the procedure. This research has been exempted from formal ethical review by the local committee (OPBA, Organismo Preposto al Benessere Animale) because providing veterinary clinical care with non-experimental purposes, in this case it falls outside the scope of the art. 2 paragraph b of Decreto Legislativo March 4, 2014, n. 26 (Implementation of Directive 2010/63 / EU on the protection of animals used for scientific purposes).

### Procedure

Before carrying out VEP testing, all animals underwent routine neurologic and ophthalmic examination as part of a general health check. Neurologic examination included observation of mentation, posture and attitude, evaluation of the cranial nerves, postural reactions and spinal reflexes. Ophthalmic examination included slit-lamp biomicroscopy, ophthalmoscopy and Schirmer Tear Test type I (STT I). The animals with neurologic abnormalities were excluded from the study. Therefore, statistical analysis was carried out exclusively on patients who did not present with ocular abnormalities upon ophthalmic examination.

VEP tests were carried out on anesthetized animals. General anaesthesia was induced and maintained by administration of isoflurane (induction was carried out by inhalation (mask) and maintained through endotracheal intubation with Isoflurane 3%). Body temperature was maintained within the normal ranges (40–41 °C) using a Shor-Line® Thermal Pad (model 712.0000.03) placed under the anesthetized animals and the temperature room was maintained at 22 °C. All tests were performed between 9 am and 12 am. The recording of the VEP tests was performed using an Electromyography and Evoked Potentials Systems (MyoHandy, Micromed, Treviso, Italy).

Animals were positioned in sternal recumbency with their heads raised by a support in order to allow correct luminous stimulation. VEPs were recorded using bipolar method, with stainless-steel needles electrodes (size 10 × 0,25 mm ) applied subcutaneously to the midline of the forehead between the eyes (Fpz, negative electrode), on the nuchal crest in the occipital region (Oz, positive electrode). The ground electrode was placed subcutaneously at Cz (vertex).

Prior to recording, no mydriatic drugs were instilled in the eyes because sufficient mydriasis was obtained under the anaesthetic plan. Between sessions recording the eyes were moistened with artificial tears (Epigel®). All recordings were made keeping the animal to the light in a quiet and foodlit room for at least 30 min allowing a proper adaptation of the eye to light.

Stimuli consisted in a flash of light of 10,000 mcd s/m$^2$ generated by a xenon lamp photo-stimulator (Flash Stimulator; Micromed®, Treviso, Italy), that produces a bright white light that closely mimics natural sunlight. The flash lamp was directly triggered by MyoHandy® device. The xenon lamp unit was located at 15 cm in front of the eye under examination and the eyelid was gently opened, while the contralateral eye was covered with a black eye patch. Two series of consecutive stimulations were carried out on each eye: the first series at a frequency of 1 Hz and the second at a frequency of 6 Hz with a 3-min interval between the two series.

The duration of the recording sessions for each animal was about 20 min. The duration of the light stimulus was approximatively 10 μs. Two waveforms were recorded from each eye, with an average of at least 200 flash responses, at both frequencies of stimulation. The two waveforms are obtained by averaging 200 raw waveforms point by point, each of which is obtained from the response to a single light flash. A double-waveform recording is commonly used to study all evoked potentials and helps to define waveform repeatability and highlight possible random peaks due to muscle artifacts. The final measurement was carried out only on the first of the two waveforms. Low and high band pass filter settings were at 0.1 Hz and 100 Hz, respectively; 50 Hz filtering was not required. The gain of the differential amplifier was 10 μV/Division.

## Data analysis
Waveforms were analysed to identify reproducible peaks from random variation of the baseline. Measurements consisted of the absolute latencies, expressed in milliseconds (ms), each of the six peaks, identified as P1, N1, P2, N2, P3 and N3 and the peak-to-peak amplitudes expressed in microvolt (μV). The measured relative latencies (interpeak) were P1–P2, P2–P3, P1–P3, P1–N2 and N2–P3, whilst the relative amplitudes of the potential considered were P1–N1, N1–P2, P2–N2, N2–P3 and P3–N3. Absolute latency was defined as the time from stimulus onset to the peak of a wave. Relative latency or interpeak was defined as the interval between two peaks. The relative amplitude was calculated as the mathematical difference between the absolute values of electrical potential between two peaks.

Positive and negative peaks latencies and potentials were measured using the cursor on the computer monitor and were recorded to the nearest 0.1 ms. The positioning of the

cursor on the monitor has been defined by the operator and the computer has supplied the values of amplitude and latency for each peak identified. In general, positive and negative peaks on the waveforms are represented by the maximum and minimum values of the curve recorded during the stimulation. Sometimes, the concept of maximum value and minimum on the curve is hardly applicable. For this reason, conventionally the slope changes of the waveforms are considered to be peaks too. The elevation changes with angle (clockwise) greater than 180° were considered positive, while the elevation changes with angle of less than 180° were considered negative.

The positive peaks (upwards) are indicated with the letter *P* and a progressive arbitrary number, and the negative peaks (downwards) are indicated with the letter *N* and a progressive arbitrary number. To meet the concept of neurophysiological repeatability of evoked potentials, each peak to be considered true had to be shown on both the waveforms with the same absolute latency from the stimulus.

Descriptive statistics consisting of mean (M), variance (Var), standard deviation (SD) and standard error (SE) for each absolute and relative latency and amplitude measurements were calculated. These are the measures that we used to define the statistical dispersion of the data, hereinafter called variability. We did not use measures of statistical dispersion. Statistical analysis was performed by pooling the data obtained from the eyes of all animals together.

## RESULTS

The results of the VEP recordings are summarized in Tables 1–6. A maximum of six peaks were identifiable in the recordings, consisting in three positive peaks (P1, P2, and P3) and three intervening negative peaks (N1, N2, N3). As regards general conformation of waveforms, a few differences were observed between species (Fig. 1). It must be pointed out that in all subjects that underwent testing, lower stimulation frequency waveforms (1 Hz) were more evident than higher stimulation frequency waveforms (6 Hz).

The statistical analysis of the results obtained showed that P1, N2 and N3 peaks are present on all recordings, whilst the remaining peaks are not always measurable. N1 was present in 73% of recordings at 1 Hz and in 55% of those at 6 Hz; P2 was present in 91% of recordings at 1 Hz and in 73% of recordings at 6 Hz and N3 in around 91% of recordings at 1 Hz and in 82% at 6 Hz. The variability of the absolute latencies of all peaks within the group was rather limited despite having considered different species of birds of prey even if of similar structure and size.

However, the analysis of the inter-peak latencies shows that at both stimulation frequencies the values that are always measurable in all waveforms are P1–P3, P1–N2 and N2–P3. These values are also those that highlight a lower variability than compared to SD and Var. The low variability of the absolute and relative peak latency values may be of clinical relevance. On the other hand, the relative amplitude of the interpeak potentials, calculated in intervals P1–N1, N1–P2, P2–N2, N2–P3 and P3–N3 is extremely variable and at present does not allow to hypothesize its use in the clinical practice.

**Table 1  Summary of absolute peak latency values expressed in msec for FVEPs at 1 Hz stimulus frequency.**

| 1 Hz | P1 | N1 | P2 | N2 | P3 | N3 |
|---|---|---|---|---|---|---|
| N | 11 | 8 | 10 | 11 | 11 | 10 |
| Mean | 12.7 | 23.0 | 26.3 | 30.3 | 37.2 | 43.0 |
| Stand. Dev. | 0.8 | 2.6 | 2.6 | 2.4 | 1.7 | 1.2 |
| Variance | 0.7 | 6.7 | 6.6 | 5.9 | 3.0 | 1.3 |
| Stand. Error | 0.2 | 0.9 | 0.8 | 0.7 | 0.5 | 0.4 |

**Table 2  Summary of absolute peak latencies values expressed in msec for FVEPs at 6 Hz stimulus frequency.**

| 6 Hz | P1 | N1 | P2 | N2 | P3 | N3 |
|---|---|---|---|---|---|---|
| N | 11 | 6 | 8 | 11 | 11 | 9 |
| Mean | 14.0 | 23.0 | 25.3 | 29.0 | 35.9 | 42.3 |
| Stand. Dev. | 0.9 | 1.6 | 2.1 | 2.5 | 1.7 | 1.7 |
| Variance | 0.9 | 2.5 | 4.6 | 6.5 | 3.0 | 3.0 |
| Stand. Error | 0.3 | 0.6 | 0.8 | 0.8 | 0.5 | 0.6 |

**Table 3  Summary of interpeak latency values expressed in msec for FVEPs at 1 Hz stimulus frequency.**

| 1 Hz | P1–P2 | P2–P3 | P1–P3 | P1–N2 | N2–P3 |
|---|---|---|---|---|---|
| N | 9 | 9 | 11 | 11 | 11 |
| Mean | 13.9 | 11.6 | 24.5 | 17.5 | 6.9 |
| Stand. Dev. | 2.2 | 1.6 | 1.8 | 2.5 | 1.4 |
| Variance | 4.7 | 2.6 | 3.4 | 6.5 | 2.1 |
| Stand. Error | 0.7 | 0.5 | 0.6 | 0.8 | 0.4 |

**Table 4  Summary of interpeak latency values expressed in msec for FVEPs at 6 Hz stimulus frequencys.**

| 6 Hz | P1–P2 | P2–P3 | P1–P3 | P1–N2 | N2–P3 |
|---|---|---|---|---|---|
| N | 7 | 7 | 11 | 11 | 11 |
| Mean | 11.6 | 10.9 | 22.0 | 15.10 | 6.9 |
| Stand. Dev. | 1.6 | 1.4 | 1.4 | 2.1 | 1.1 |
| Variance | 2.5 | 1.8 | 2.0 | 4.6 | 1.2 |
| Stand. Error | 0.6 | 0.5 | 0.4 | 0.6 | 0.3 |

In some case there is a tendency of the first (P1) and second (P2) positive wave to overlap. The frequency of overlap between P1 and P2 was of 9% with a stimulation at 1 Hz and of 27% with a stimulation at 6 Hz. Under these conditions, the N1 and P2 values were not considered in statistical analysis due to their difficult localization. Finally, it was not possible to define all N3 values due to reduced amplitudes and lack of repeatability in the control waveforms.

**Table 5** Summary of interpeak amplitudes values expressed in µV for FVEPs at 1 Hz stimulus frequency.

| 1 Hz | P1–N1 | N1–P2 | P2–N2 | N2–P3 | P3–N3 |
|------|-------|-------|-------|-------|-------|
| Mean | 69.2 | 10.8 | 14.1 | 12.9 | 3.7 |
| Stand. Dev. | 25.1 | 8.0 | 9.4 | 10.2 | 2.5 |

**Table 6** Summary of interpeak amplitudes values expressed in µV for FVEPs at 6 Hz stimulus frequency.

| 6 Hz | P1–N1 | N1–P2 | P2–N2 | N2–P3 | P3–N3 |
|------|-------|-------|-------|-------|-------|
| Mean | 72.2 | 20.1 | 8.4 | 18.7 | 10.4 |
| Stand. Dev. | 30.8 | 16.3 | 6.3 | 8.9 | 8.7 |

As well as the previously described peaks, on the recordings of all the birds of prey studied, as regards to amplitude of potentials, delayed, unrepeatable and less evident peaks were also highlighted between P1 and N3.

## DISCUSSION

The results of this study have shown that it is possible to record FVEPs in diurnal birds of prey using the technique that has already been described and used in dogs (*Strain, Jackson & Tedford*, *1990*). The main feature of this electrodes setup is to obtain waveforms composed by both the potentials generated by the ocular structures and both from those produced by the visual pathways. This is due to the bipolar configuration, in which both electrodes are exploring. Moreover, the morphology of the waveforms achieved and the peaks of the potentials considered (P1, P2 e P3) are the same as those observed in dogs. Therefore, the use of the use of VEPs in clinical practice can be hypothesized in the future. In fact, the absolute and relative latency values of the FVEPs in the birds of prey studied proved to have reduced variability.

The clinical usefulness of FVEPs is well known in human medicine, as it allows to objectively assess the functional integrity of the visual pathways, from the retina to the visual cortex, even during general anaesthesia and coma, or when carried out on neonates (*Chiappa & Hill*, *1997*). FVEPs are commonly used in clinical practice also in dogs to study visual function but with some differences compared to man. In fact, in animals, FVEPs require use of anaesthetic drugs due to the lack of active collaboration during the test. Compared to what is described in man, where the test is normally carried out on awake individuals and a broad inter-individual variability of potentials exists (*Odom et al.*, *2010*), in diurnal birds of prey, as is the case in dogs, the use of a general anaesthesia reduces the variability of the evoked visual responses (*Kimotsuki et al.*, *2005b*).

Two further elements play a role in determining the usefulness of this protocol in birds of prey: the first is adaptation of the eye to light. In fact, the test is carried out in phototopic conditions, which determine a retinal potential that has reduced amplitude and duration, which gives the possibility of better highlighting the potential delay produced by

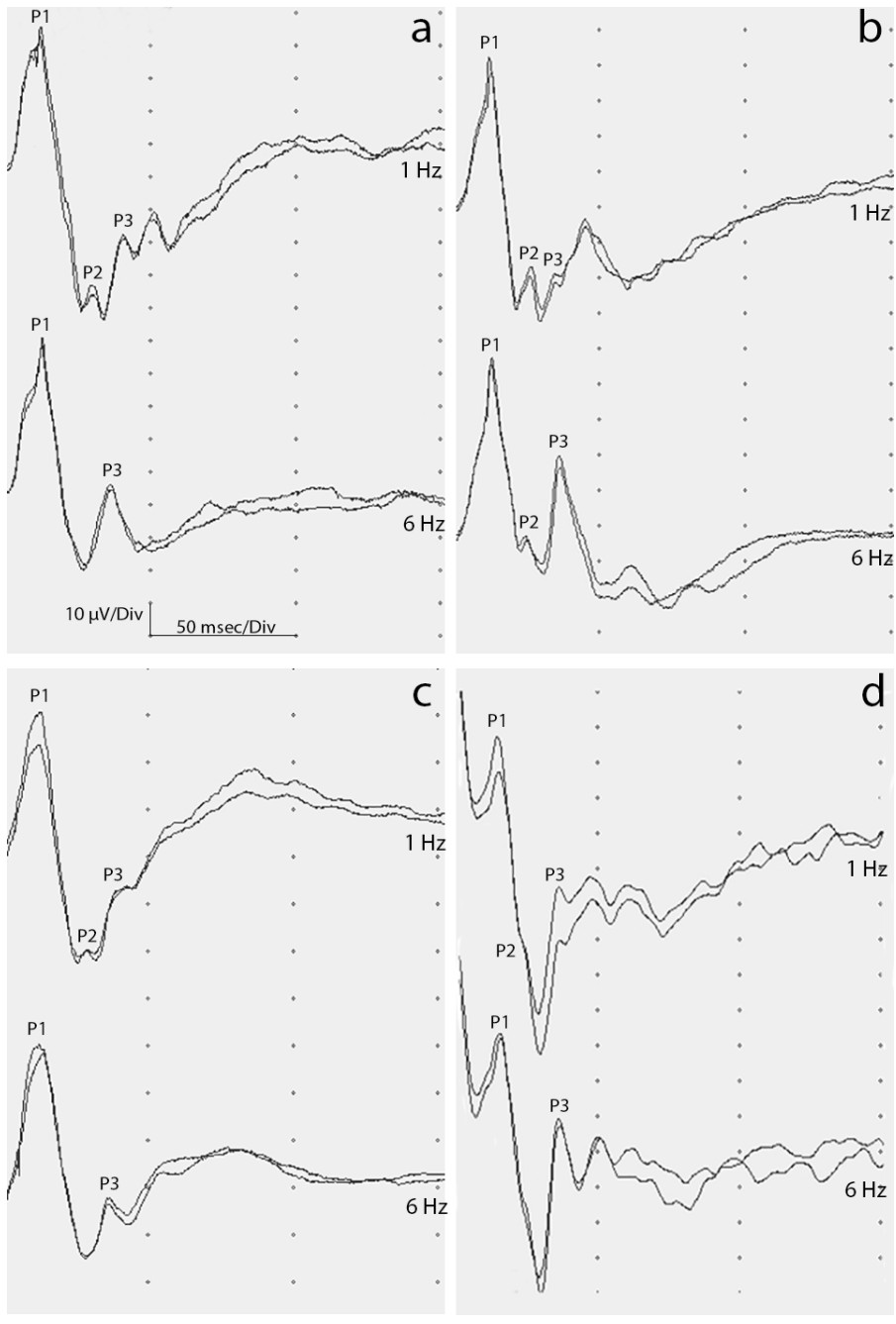

**Figure 1  Images of FVEP waveforms.**

the post-retinal nervous structures. The second element is the positioning of the electrodes that being both active and given the small size of the skull of the birds of prey allow to determine the potentials produced by the eye and by the entire visual path on the same waveform.

   In dogs, the usefulness of the FVEP test is related to the correlation between the function of particular structures of the visual path and the presence of precise peaks

on recordings. In particular, it is possible to identify the neuro-anatomical location of the visual lesions following a lack of determined potentials or following an increase in their latency times. In fact, Sims demonstrated the correspondence between ERG b wave and FVEPs P1 wave in the dogs, and also that complete lesions of the optic nerve cause the disappearance of all potentials after N1 (*Sims et al.*, *1989*). Kimotsuki, again in dogs, showed that a lesion of the Lateral Geniculate Body causes the immediate disappearance of peaks N2 and P3 (*Kimotsuki et al.*, *2005b*). Therefore, it is possible to state that P1 represents retinal potential and can be identified with wave B of the ERG; the N1–P2 interval is generated by the optic nerve, by the chiasm and the visual pathway; interval N2–P3 is generated by the lateral geniculate body and by optic radiations.

In birds of prey, similar conclusions are not possible due to lack of accurate data on their neuro-functional anatomy, even if significant neuroanatomical similarities with the visual pathways of mammals exist and could allow to make parallel hypotheses. In fact, in birds of prey, there are two parallel visual pathways: the tecto-fugal and thalamo-fugal pathways. The first pathway corresponds to the extra-geniculostriate system in mammals and in particular in primates, whilst the second pathway corresponds to the geniculostriate system. The tecto-fugal pathway (collothalamic) is composed of axons of the optic nerve that intersect with different percentages according to the species to form the optic chiasm. Then, these fibres reach the optic tectum and then the round nucleus of the thalamus and finally, the ectostriatum nuclei. The ectostriatum is a wide longitudinal cerebral structure incorporated in the dorsal ventricular ridge (DVR) that is mainly responsible for the elaboration of diurnal sight, whilst the lemnothalamic pathway goes from the retina to the dorsal thalamic nuclei and ends on the visual cortex that in birds is called Wulst. The first pathway (collothalamic) is more developed in species with eyes located laterally (ground-feeding birds), the second one (lemnothalamic) is more developed in owls and hawks for processing the frontal binocular field (*Shimitsu & Bowers*, *1999*; *Husband & Shimizu*, *2001*).

The results of this study, also very limited for the small number of animals used, indicates that is possible to obtain FVEPs by birds of prey. The hypothesis of a clinical use of this test for the functional evaluation of the visual pathways, envisages further studies with a larger number of animals. It will also be necessary to clarify the neuro-anatomical origin of these evoked potentials in the various species considered.

### Funding

The authors received no funding for this work.

### Competing Interests

The authors declare there are no competing interests.

## Author Contributions

- Maurizio Dondi conceived and designed the experiments, performed the experiments, analyzed the data, wrote the paper, prepared figures and/or tables, reviewed drafts of the paper.
- Fabio Biaggi conceived and designed the experiments, performed the experiments, analyzed the data, wrote the paper, prepared figures and/or tables.
- Francesco Di Ianni contributed reagents/materials/analysis tools.
- Pier Luigi Dodi performed the experiments.
- Fausto Quintavalla contributed reagents/materials/analysis tools, reviewed drafts of the paper.

## Animal Ethics

The following information was supplied relating to ethical approvals (i.e., approving body and any reference numbers):

Organismo Preposto al Benessere Degli Animali (D. R. n. 350 Reg. LII del 01 agosto 2014): PROT.N. 34/OPBA/2016.

## Data Availability

The raw data has been supplied as Supplemental Dataset.

## Supplemental Information

Supplemental information for this article can be found online at http://dx.doi.org/10.7717/peerj.2217#supplemental-information.

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
