# Peer review of "Flash visual evoked potentials in diurnal birds of prey"

_PeerJ, doi:10.7717/peerj.2217_

## Round 0.1 · original submission · Major Revisions

· Academic Editor

Major Revisions

The manuscript is of interest due to the data and methodology provided for VEP studies in birds of prey,

The reviewers have been very specific in regards to their recommendations and in most cases are in close agreement and I encourage the authors to take these excellent suggestions into account to improve the quality of the manuscript..

The low "n" that is of concern to reviewer 2 can be addressed as suggested by reviewer 1: emphasizing the pilot nature of this study and including discussion of future opportunities enabled by the present work.

Reviewer 1 ·

Basic reporting

Introduction
1) It is clear from the introduction that the authors attempt to place their research into a larger narrative. However very little is provided in background studies into the use of VEPS or electroretinograms (ERG) in avian species.
2) Please clarify how exactly VEPs generated in response to simple flash stimuli can be used to probe retinal and central function in birds. You do not provide enough evidence in support of applying VEP recordings to your birds. Rather you only briefly mention optic nerve lesions and central pathways (line 77). What specifically did these lesion studies show in terms of VEP responses – do the VEPS disappear completely or did they change in reproducible ways (i.e. lesion of visual cortex removed only later peaks and kept initial peaks untouched. This information will help you to sell the method of VEPS as a diagnostic tool which I think you need to do in a pilot study such as this.

Minor Points
Abstract and Introduction
Line 23. What are the numbers n.2 referring to? Is this the number of animals? Or the numbers of eyes? Can you place the numbers in parentheses (N=2) after you mention the species.
Line 26 poor expression ‘random variation of baseline’. Consider changing to commonly used expression ‘background noise’ or ‘resting electrical activity’. The point here is that it’s not evoked activity – you haven’t shown randomness
Line 30 What other animal species do you refer to? – later on in the manuscript you compare the dog – which is not a bird of prey. Can you be more specific in what other animals species. Better yet, can you cite avian papers.
Line 40 ‘certain pathologies’ is very vague term. Please be specific
Line 42 towards what species? And what sort of veterinary interest – clinical/ scientific/ anecdotal?
Line 46 what ‘eye pathologies’ are they traumatic eye injuries? Are they anatomical pathologies or functional pathologies? Please specify.
Line 49 what centres are you referring to? The bird of prey specific centres or the Murphy 1987 / Labelle 2012 papers?
Line 50 …. Of animals is sent… poor expression
Line 56 spelling mistake – Wulst, also why is wulst and entopallium capitalised?
Line 59 …as it does not allow observers? …. Who is not being allowed to identify? Please clarify
Line 63 remove the word ‘us’ as it is not an appropriate pronoun
Line 63 identify ‘alterations’ – vague term – what alterations? Structural or functional?
Line 66 why capitalised cranial nerve examination (CNE?). please change
Line 72 who recommends this test? Clinical Practice? Or Patient demand? Please specify who recommends these tests.
Line 80-83 confusing sentences. What are tracks? My definition of tracks as an electrophysiologist is completely different to how you’re using this term. Do you mean traces? Or waveforms? Not tracks. Please change.
Line 86 unusual placement of reference after words pilot study. Please change location of reference to end of sentence.

Experimental design

Major Points
Methods
Please indicate clearly in your methods section if Animal Ethics Approval was required or why your research is exempted from formal ethical review by your local committee.

1) Your methods section is missing many critical pieces of information. You are arguing that your study is a pilot study and therefore you need to clearly outline all pieces of equipment and the methods that you’ve employed in your recordings.
2) How did you confirm the position of your electrodes at all locations (including Oz and Cz regions). Why did you pick these regions for electrode placement? Are there stereotaxic features that you can use to ensure electrode placement? Or are there externally derived anatomical landmarks?
3) What time did you make your recordings? There is evidence that in some bird species (Wu et al., 2000 J Biol Rhythms 15(4):317) that flashed evoked potentials change throughout the day. Please comment on the time of day recordings were made and potential impact in diurnal birds.
4) What recording method did you use? Are you recording single ended potentials (Cz relative to ground) or are you recording bipolar/differential potentials (Oz-Cz relative to ground). Please clarify.
5) You mention that your recordings were made with prior adaptation to the light. What light are you referring to? Please give specific details on the amount of light the visual system was presented with during adaptation, furthermore – how long was adaptation process for? Please provide information.
6) What was the gain on your amplifier? You mention filter settings but fail to tell us amplification or gain used. Also what amplifier did you use in your recording apparatus?
7) Please provide more information on the intensity of the stimulus – how was the light source calibrated, what is the spectrum of the stimulus used? What is the rise/fall and pulse duration of the stimulus?
8) How did you trigger your stimulus and recording system – was it controlled externally by a pulse generator or did the MyoHandy send a trigger pulse to the photo-stimulator.
9) Why have you chosen 1Hz and 6Hz stimulus frequencies? Is there a reason for using these two frequencies? Did you hypothesize any stimulus-frequency dependent change in VEP waveform? Please comment.
10) Please comment on how you undertook peak detection in your response waveforms, this process underpins your statistical analysis (absence / presence of peaks) and thus required more information as to the methods of peak detection.
11) Averaging – you mention that your VEPS were generated in response to 200 pulses of light, was each individual pulse of light averaged to create 1 recording? Please make clear the method of averaging.

Minor Points
Line 93 please change location of the number of animals to after species mention – they are confusing. Also – how many Left and Right eyes did you measure from?
Line 100 please outline what ‘neurologic’ components of your examination consists of
Line 102 I know they’re patients to you – but please change the word patients to animals or birds with abnormalities - how were these abnormalities assessed clinically?
Lines 108-9 poor expression ‘Thanks to a heating pad’ – please change expression and provide details on temperature of the pad and manufacturer of pad. Please provide details on the approximate temperature of the room (electrophysiological responses tend to slow down (get wider) and smaller in amplitude as organisms cool down.
Line 109 what tests? You are not clear in telling the reader what exactly your MyoHandy box does. Does it control the stimulus (flash) or simply record EEG/EMG data?
Line 112 what size were the electrode needles used
Line 124 was the eye moistened or closed between your recording sessions? Please clarify how long the recording sessions took.
Line 128 please confirm filter settings and the type of filters used. Were they software or hardware defined. Also – please correct your figure legend or your methods section. You mention 2 different low filter numbers (1Hz in methods, 0.1Hz in legend).
Line 140 please clarify what you mean by ‘followed by a number’- are these arbitrarily assigned or do they simply increment with each successive negative / positive deflection like they do in other VEP recordings.
Line 139-40 your VEP traces were recorded by MyoHandy machine and then analysed by moving a cursor on a monitor? Can you please clarify if the cursor gave a direct readout (i.e. user defined location but computer defined readout) or if an observer estimated the response magnitude.
Line 146 please give some more detail on qualitative analysis – did this involve overlaying responses from 2 or more species? Or simply counting number of peaks in the response.

Validity of the findings

Major Points
Results
1) You do not provide any evidence that your responses reflect genuine visually evoked potentials (VEPS). It is possible that your responses are simply artefacts from the visual stimulus or reflect a EMG. I would suggest providing data where you have blocked the stimulus from the animal and recorded the response. If truly visually evoked – blocking the stimulus should reveal no waveforms. Alternatively, you can undertake stimulus intensity/ amplitude functions (where you turn down the stimulus intensity) and measure the amplitudes of the peaks. If the amplitudes do not change as the stimulus turns down – then they’re not dependent on the stimulus and thus represent an artefact.
2) I am unsure if you have pooled the data from all your animals together in your analysis. Can you please clarify if your dealing with each animal species separately or if you have pooled the data from all eyes together?
3) The way you deal with ‘variability’ is very confusing. Please provide a statistical method or alternative way of clearly defining what you mean by variability. In discussing variability of amplitudes are you referring to a coefficient of variation (CV) or some other measure (simply standard error or standard deviation). With your handling of the latencies – are you referring to response jitter (i.e. the time at which the peak occurred across all responses) or something entirely different. Please clarify what you mean by variability.

Figures
1) I would strongly urge you to include a at least 50-100 ms of baseline (non-evoked) recordings in your raw traces. Your reader needs to see what resting noise you have in your recordings as you are comparing visual-evoked responses to the resting background noise
2) Please remove one trace or at least clearly define what the second trace refers to. This is quite confusing.
3) What are the upper trace pairs? This is not clear – do the upper traces refer to 1Hz or 6Hz or left eye or right eye? Please define or annotate the figures. Alternatively - split figure into 1Hz and 6Hz traces for clarity.

Figure 1 legend–
(i) you mention Low Band Pass filter set at 0.1 Hz – this conflicts with your methods section. Also – was it a simple Low Pass Filter? or a Band Pass filter – these are two separate things, if you applied a band pass filter please give the pass frequency range. In fact this information is not required in the legend if you correctly place it in your methods section. Please fix.
(ii) what is a sweep? No mention anywhere else in the text what a sweep is. Please remove or define.

Tables:
Please confirm that the N in the tables refer to the number of animals or the number of eyes recorded from

Please consider changing the order of the tables – I think the amplitude measurements should come first (Table 5 should be Table 1). Amplitudes can be used to argue that you are able to generate a response in your animals and opens up your latency analyses. .

Discussion -
(1) You repeatedly place your results into the context by referring to canine VEP responses. This largely inappropriate due to vast anatomical differences that exist between avian and canine retina as well as central visual pathways. If no data derived from avian models exist, please mention this and then critically analyse how your VEP recordings may potentially differ from other species (e.g. what anatomical differences exist, what neurophysiological differences exist) and thus reduce your ability to interpret the responses.
(2) Please discuss the likely effects of anaesthesia on your VEP responses, what effects are known on the impact of isoflurane (or other anaesthetic agents) on recording population responses from in vivo models.
(3) With regards to the contribution of retinal and post-retinal structures to your recordings there is data from the mouse (which has a very small cranium) investigating the effect of changing electrode locations on ERG and VEP contributions to the recorded potentials (figure 3 in Ridder et al., (2006) Vision Research (46):902-913). Please comment on how your data may reflect retinal and central structures.
(4) You do not discuss why there were differences between the 1Hz and 6Hz stimuli frequencies. Are these birds able to response to 6Hz stimuli? Please include a discussion on possible mechanisms underlying 1 and 6Hz. Additionally, please justify why you are using 1 and 6Hz stimuli is there a particular reason for these 2 stimuli frequencies?
(5) I think you are missing some comment on future studies that will expand on your broad findings (that VEPS can be recorded from these animals). Alternatively- I think will need to propose (at least briefly) a protocol on how to reproduce your experimental methods. It is not enough to state that FVEPS are clinically indicated.

Minor Points
Results
Line 152 please clarify what ‘track morphology’ means – confusing sentence
Line 157 please clarify how you determined whether a peak was not always measurable
Line 162 ‘…it emerges that…’ wrong tense
Line 169 what do you mean ‘from a visual analysis’ isn’t this what you’ve been doing? Please clarify this expression.
Line 170-72confusing sentence: These overlapping waves… please clarify what you’re referring to.
Line 176 what are control recordings? You have not mentioned control conditions or provided any data under ‘control’ experiments.
Line 179 please remove or change the final sentence. It is vague and adds nothing to your results section.
Discussion
Line 187 ‘…in clinical practice can be hypothesized.’ You mention a protocol in this sentence and argue that it can be used. Yet you do not provide any protocol data in your methods of sufficient quality for a trained scientist to repeat your study. Please either change this sentence, or provide a meaningful protocol for others to follow.
Line 191 ‘collaborate’ – poor choice of word, please change
Line 194 ‘pharmacological containment’ – poor expression, please change
Line 213-16 in what species are you referring to when you state that P1 represents retinal potential etc. Are you talking about dogs or your birds of prey? Please clarify
Line 232 remove the apostrophe – it’s, replace with it is.

Additional comments

Thank you for allowing me to review the manuscript by Dondi et al., I would like to commend the authors on undertaking a truly interesting set of experiments. From my reading of the manuscript the authors have attempted to catalogue some gross responses of the central visual pathways in birds of prey by recording evoked potentials in response to simple visual stimuli in anesthetised animals. While this topic is of interest from a comparative aspect, as well as being clinically relevant, the paper as it currently stands is fails to provide sufficient information to convince this reader to accept their arguments. Furthermore, the authors position the manuscript as a pilot study and hint at the clinical usefulness of their particular techniques however fail to provide sufficient detail on key methodological and mechanistic domains

Reviewer 2 ·

Basic reporting

No Comments

Experimental design

No Comments

Validity of the findings

No Comments

Additional comments

It is interesting to see a VEP study in birds of prey. Here are my comments:
1. It is known that different strains of animals can show very different VEP traces, therefore the sample size (n=6 in total and only n=1 or 2 for each raptor strain ) presented here by the authors was too small and impossible for any statical analysis. Ideally, one eye should be selected at random for analysis (Armstrong, Ophthalmic Physiol Opt 2013).
2.The disadvantage of using isoflurane in VEP studies is that it can significantly suppress the EEG signals (Makela et al.; Electroencephalogr Clin Neurophysiol 1996)(Jehle et al Doc Ophthalmol 2009) - this needs to be discussed in the Discussion section. Also, needle electrodes tend to provide more variable and smaller VEP signals (You et al. Doc Ophthalmol 2011).
3. Is there a reason that the authors used 1 Hz and 6 Hz stimuli frequencies for this study? Instead, different light intensities are sometimes used in lab VEP studies to separate the rod and rod-cone mixed pathways. Can the authors provide the strength of flash stimulus used in this study in cd s/m-2?
3.0 cd s/m-2 is the ISCEV standard (Odom et al Doc Ophthalmol 2010).
4. My major concern:
The latency of the major peak (P1) appears to be too early (~10-15 ms). The ERG a wave implicit time was about 15 ms in raptors (Labelle et al. Vet Ophthalmol 2012). We don't expect to see a VEP signal earlier than ERG. The authors are showing double traces in each panel in Fig 1. Are they from right and left eyes respectively? If yes, they looked identical.

---

## Round 0.2 · Minor Revisions

· Academic Editor

Minor Revisions

Please make the minor changes recommended by the reviewer under the section "Basic reporting"

Although not ideal, I accept your rebuttal of the control responses.
If possible, please provide better traces for the VEP recordings that show the background levels prior to initiation of P1.

Reviewer 1 ·

Basic reporting

This report is in english that is understandable to an educated audience
The new introduction is of sufficient depth and links previous research well into the current research question.

Line 83 and Line 92: ...'Searching the veterinary....' are repetitious. Consider removing one instance.
Line 152 - the word software is superfluous in this context
line 181- please change the sentence or the word order paying particular attention to the word 'dimensionless'... the sentence as it currently stands is not appropriate

Line 188 - lower stimulation frequency waves (1Hz) were more evident... I think you're referring to 'waveforms' and not 'waves'. Please consider changing to correct terminology (waveforms).

Throughout the manuscript the authors continue to use the word 'Tracks' (see line 217)- I remain steadfast in my argument that this word is not appropriate in electrophysiological data of this type. I would strongly urge you to consider using the word 'waveforms'.

Experimental design

No comments - sufficient information is presented for other scientists to repeat this set of experiments.
Experimental protocols are appropriate for the research question at hand

Validity of the findings

I would still like some data on if these responses were in direct result of photic stimulation (rather than electrical field stimulation from the hand held flash bulb discharge).

I disagree with your rebuttal that control responses (i.e. looking at VEPS without photic stimulation) are not possible. One protocol to achieve this 'control' response is simple. You cover the eye (or keep it closed) (Is it possible to temporarily place a leather hood on the birds during testing?) during stimulation and you record the responses. The responses (VEP) waveforms should approximate the average noise of the recording system.

Figures - a small section of waveform before stimulation (10-15ms) would greatly indicate the start of the VEP response. I think your figure 1 waveforms appear to start halfway up the initial response (before P1) rather than starting from a true resting baseline. Please consider starting your waveforms before the VEP response.

Your discussion is adequate.

Additional comments

This remains of interest to the larger scientific community. I understand that this data is likely derived from 2013 (approximately) and so some of the requested data is likely not available (i.e. VEP responses when the eye is shut). I argue that this set of data is critical in establishing that your 'tracks' (or what I would call 'waveforms') are indeed from photic stimulation (and not an electrical artefact).

Thank you - good luck for future scientific endeavours.

---

## Round 0.3 · accepted · Accept

· Academic Editor

Accept

Thank you for addressing the editor's and reviewer's comments and I am pleased to accept your paper in PeerJ.